# Drivers of species knowledge across the tree of life

**Stefano Mammola**[1,2,3*], **Martino Adamo**[3,4], **Dragan Antić**[5], **Jacopo Calevo**[6,7], **Tommaso Cancellario**[1], **Pedro Cardoso**[2], **Dan Chamberlain**[4], **Matteo Chialva**[3,4], **Furkan Durucan**[8], **Diego Fontaneto**[1,3], **Duarte Goncalves**[9], **Alejandro Martínez**[1], **Luca Santini**[10], **Iñigo Rubio-Lopez**[1], **Ronaldo Sousa**[11], **David Villegas-Rios**[12], **Aida Verdes**[13], **Ricardo A Correia**[14,15,16,17]

[1]Molecular Ecology Group (MEG), Water Research Institute (CNR-IRSA), National Research Council, Verbania, Italy; [2]Laboratory for Integrative Biodiversity Research (LIBRe), Finnish Museum of Natural History (LUOMUS), University of Helsinki, Helsinki, Finland; [3]National Biodiversity Future Center, Palermo, Italy; [4]Department of Life Sciences and Systems Biology, University of Turin, Torino, Italy; [5]University of Belgrade - Faculty of Biology, Belgrade, Serbia; [6]Royal Botanic Gardens, London, United Kingdom; [7]School of Molecular and Life Sciences, Curtin University, Perth, Australia; [8]Department of Aquaculture, Isparta University of Applied Sciences, Isparta, Turkey; [9]CIIMAR, Interdisciplinary Centre of Marine and Environmental Research, University of Porto, Matosinhos, Portugal; [10]Department of Biology and Biotechnologies "Charles Darwin", Sapienza University of Rome, Rome, Italy; [11]CBMA – Centre of Molecular and Environmental Biology, Department of Biology, University of Minho, Minho, Portugal; [12]Instituto de Investigaciones Marinas, CSIC, Eduardo Cabello, Vigo, Spain; [13]Department of Biodiversity and Evolutionary Biology, Museo Nacional de Ciencias Naturales, Madrid, Spain; [14]Helsinki Lab of Interdisciplinary Conservation Science (HELICS), Department of Geosciences and Geography, University of Helsinki, Helsinki, Finland; [15]Helsinki Institute of Sustainability Science (HELSUS), University of Helsinki, Helsinki, Finland; [16]CESAM – Centre for Environmental and Marine Studies, University of Aveiro, Aveiro, Portugal; [17]Biodiversity Unit, University of Turku, Turku, Finland

**\*For correspondence:**
stefanomammola@gmail.com

**Abstract** Knowledge of biodiversity is unevenly distributed across the Tree of Life. In the long run, such disparity in awareness unbalances our understanding of life on Earth, influencing policy decisions and the allocation of research and conservation funding. We investigated how humans accumulate knowledge of biodiversity by searching for consistent relationships between scientific (number of publications) and societal (number of views in Wikipedia) interest, and species-level morphological, ecological, and sociocultural factors. Across a random selection of 3019 species spanning 29 Phyla/Divisions, we show that sociocultural factors are the most important correlates of scientific and societal interest in biodiversity, including the fact that a species is useful or harmful to humans, has a common name, and is listed in the International Union for Conservation of Nature Red List. Furthermore, large-bodied, broadly distributed, and taxonomically unique species receive more scientific and societal attention, whereas colorfulness and phylogenetic proximity to humans correlate exclusively with societal attention. These results highlight a favoritism toward limited branches of the Tree of Life, and that scientific and societal priorities in biodiversity research broadly align. This suggests that we may be missing out on key species in our research and conservation agenda simply because they are not on our cultural radar.

## eLife assessment

With a carefully collected dataset and **compelling** analyses, this **fundamental** manuscript demonstrates detailed links between societal and academic interest and natural species across the globe. In doing so, the authors reveal biases that may be diminishing our abilities to care for the species on our planet that may need our care the most. While some parts of this manuscript reflect previously published work, the authors are commended for putting all the puzzle pieces together for the first time. Their work highlights our uneven knowledge of biodiversity and its potential causes.

## Introduction

Human relationships with biodiversity trace back to our dawn as a species (*Wilson, 1993*). Wildlife permeates art, myths, and traditions; it constitutes an irreplaceable source of food and goods; and, even in the digital age, it remains one of the most powerful triggers of human emotions (*Correia and Mammola, 2023*; *Hicks and Stewart, 2018*; *Jacobs, 2012*; *Soga and Gaston, 2016*). Furthermore, the birth of modern science has turned biodiversity into a subject of intense investigation. However, scientific and societal attention toward biodiversity is unevenly distributed across the branches of the Tree of Life (*Wilson et al., 2007*). Whether for utilitarian reasons or due to conflictual emotional stimuli (*Nyhus, 2016*), we have better knowledge of some species than others (*Jarić et al., 2022*).

Widespread evidence indicates that biodiversity research has concentrated on certain lineages, habitats, and geographic regions over others (*Clark and May, 2002*; *García-Roselló et al., 2023*; *Hortal et al., 2015*; *Mammola et al., 2023*; *Šmíd, 2022*; *Troudet et al., 2017*). At the species level, for example, research interests and conservation efforts are often skewed toward vertebrates rather than other animals (*Cardoso et al., 2011a*; *Cardoso et al., 2011b*; *Leather, 2013*), plants (*Adamo et al., 2022*; *Balding and Williams, 2016*), or fungi (*Gonçalves et al., 2021*; *Oyanedel et al., 2022*).

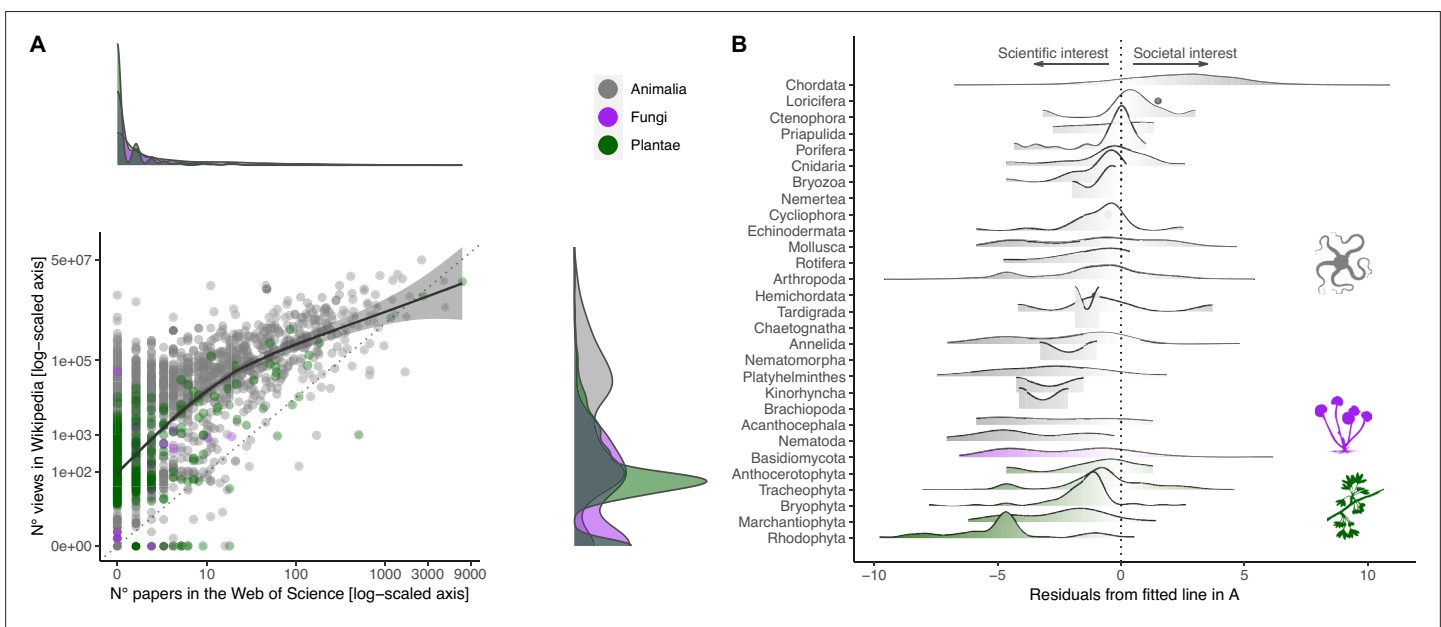

**Figure 1.** Relationship between societal and scientific interest across the eukaryotic Tree of Life. (**A**) Relationship between number of views in Wikipedia (popular interest) and number of papers in the Web of Science (scientific interest) for each species. Both axes are log-scaled to ease visualization. Density functions are provided for both scientific (above scatter plot) and societal interests (right of scatter plot) to illustrate the distribution of values. Color coding refers to the three realms of Animalia, Fungi, and Plantae. The regression line is obtained by fitting a Gaussian generalized additive model through the data ($F_{1,3017} = 2497.5$; $p<0.001$). The farther away a dot is from the fitted line, the more the attention is unbalanced toward either scientific (negative residuals) or societal interest (positive residuals). (**B**) Distribution of negative and positive residuals (from the regression line in **A**) across the species sampled for each Phylum/Division. Phyla/Divisions with only one sampled species are represented with dots.

The online version of this article includes the following figure supplement(s) for figure 1:

**Figure supplement 1.** Global distribution of the sampled species.

**Figure supplement 2.** Breakdown of scientific (**A**) and societal (**B**) interest by phylum, as well as the number of sampled species (**C**).

Furthermore, scientific and societal attention towards species may correlate, to some degree, with aesthetic features (*Adamo et al., 2021*; *Borgi et al., 2014*; *Ward et al., 1998*), online popularity (*Correia et al., 2016*; *Mammola et al., 2020*), and phylogenetic proximity to humans (*Miralles et al., 2019*), although the relative importance of these factors is likely to vary across cultural settings and societal groups. Indeed, even the selection of model organisms is not always based on functional criteria (e.g., ease of growth under controlled conditions, cell size, genome size, ploidy level; *Hedges, 2002*) and instead may be driven by economic, affective, cultural, or other subjective attributes (*Dietrich et al., 2020*).

Importantly, most attempts to quantify which features make species attractive to humans have focused on vertebrates—typically mammals and birds (*Santangeli et al., 2023*; *Haukka et al., 2023*; *Miralles et al., 2019*). This means we now possess a growing understanding of research biases for selected taxa (*Guedes et al., 2023*; *Šmíd, 2022*; *Sumner et al., 2018*; *Zvaríková et al., 2021*), but we still lack a comprehensive picture of cross-taxa features that could drive human interest in biodiversity. Here, we explored research and societal interest in organisms across the Tree of Life, asking two general questions: What are the species-level and cultural drivers of scientific interest throughout the Tree of Life? And, how do those drivers differ from those explaining societal interest? To this end, we randomly sampled 3019 species spanning 29 Phyla and Divisions (*Figure 1*; *Figure 1—figure supplement 1*). We sourced the number of scientific papers focusing on each species as a measure of scientific interest (*Figure 1—figure supplement 2A*), and the number of views of the Wikipedia page of each species as a measure of societal interest (*Figure 1—figure supplement 2B*). Furthermore, we collected species-level traits referring to morphology and ecology (size, coloration, range size, biome, and taxonomic uniqueness) and cultural factors reflecting how humans perceive and interact with biodiversity (usefulness and harmfulness for humans, presence of a common name in English, phylogenetic distance to humans, International Union for Conservation of Nature [IUCN] conservation status).

## Results

The number of scientific papers focusing on these randomly selected species varied by four orders of magnitude and showed a highly skewed distribution (*Figure 1A*). While 52% of species lacked scientific papers associated with their scientific name in the Web of Science (median ± SE = 0 ± 3.96), there was a long tail of comparatively few species attracting substantial scientific attention (the most studied species in our selection, *Ginkgo biloba* L., appeared in as many as 7280 scientific papers) (*Figure 1—figure supplement 2A*). In contrast, the distribution of the number of views in Wikipedia was less skewed (*Figure 1A*), but there was enormous disparity in societal attention across species (266 ± 25,217; range = 0–50,727,745) (*Figure 1—figure supplement 2B*). With the notable exception of Chordata (the Phylum encompassing all vertebrates), most species from other taxonomic groups attracted more scientific interest than expected from societal attention (*Figure 1B*). The few species that attracted disproportionately more societal than scientific attention were colorful, of larger size, and possessed a common name (*Figure 2*).

Next, we modeled scientific and societal interest in relation to species-level traits and cultural features using generalized linear mixed effects models, controlling for phylogenetic and geographic effects. This analysis revealed a set of drivers that were associated with a high scientific and societal interest (*Figure 3A*; see 'Materials and methods' for driver-specific hypotheses), with scientific and societal priorities largely mirroring each other. First, larger species were more attractive to both scientists and the general public. Second, species with broader geographic distributions and taxonomically unique species (i.e., with fewer congenerics) all received greater scientific and societal attention. Third, several cultural features strongly correlated with both scientific and societal interest, including the presence of a common name, whether a species is useful and/or harmful for humans, and whether a species had been assessed in the IUCN Red List of Threatened Species. Finally, there were three traits uniquely associated with societal interest in organisms: colorful species, freshwater-dwelling species, and species phylogenetically closer to humans all received greater societal attention.

Overall, both models explained ~60% of variance, with an additional ~20% captured by random effects related to taxonomic relatedness and geographic provenance. Using variance partitioning analysis, we compared the relative contribution of morphological, ecological, and cultural factors in determining the observed pattern of research and societal attention. Cultural features were the

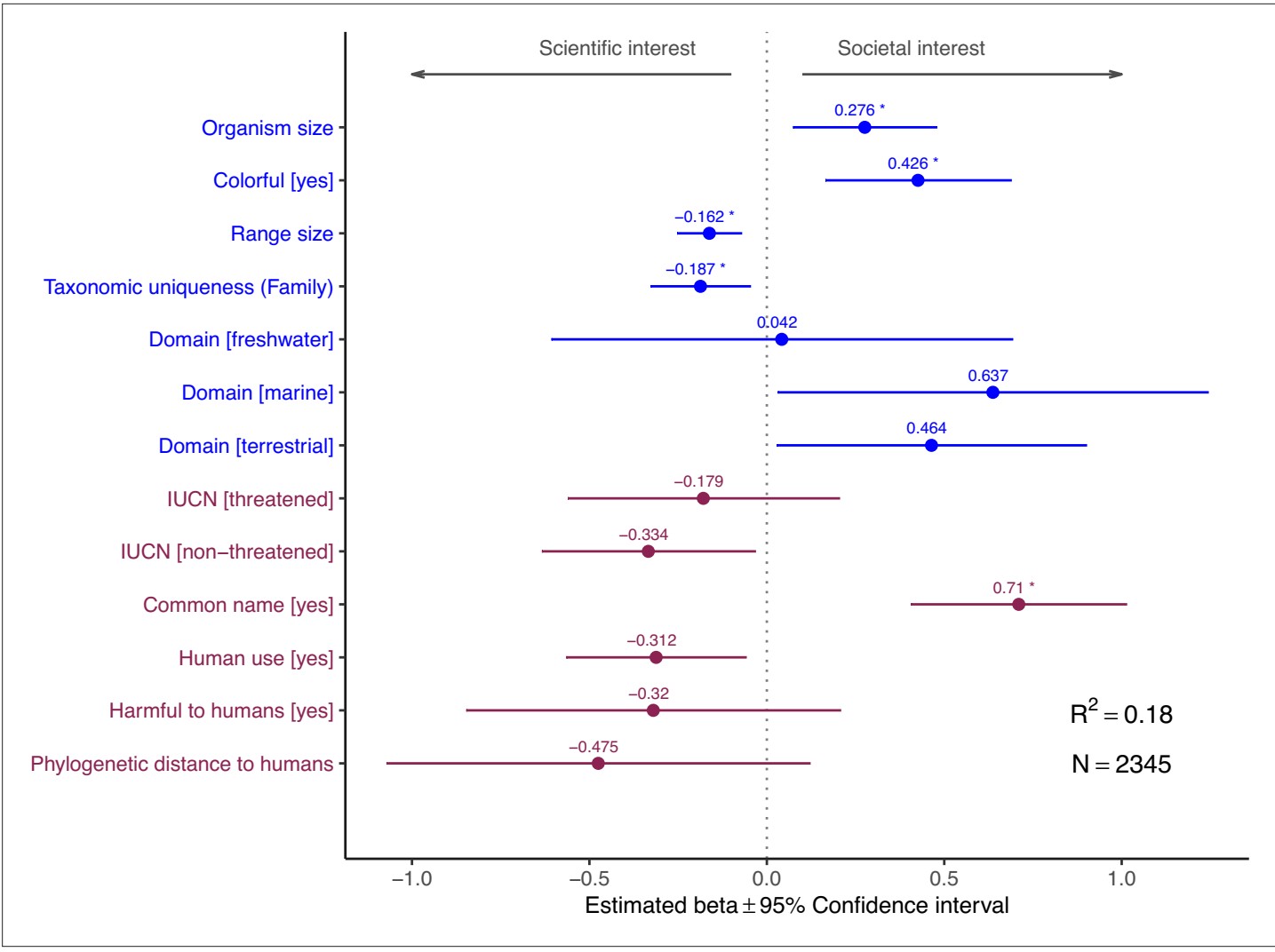

**Figure 2.** Influence of species-level traits (blue) and cultural factors (red) on relative scientific and societal interest for different taxa. Forest plots summarize the estimated parameters based on Gaussian linear mixed models testing the relationship between residuals from the regression line in *Figure 1A* and species-level traits. Positive residuals indicate species with a greater popular than scientific interest, residuals close to zero indicate species with a balanced scientific and societal interest, and negative residuals indicate species with a greater scientific than societal interest (*Figure 1B*). Baseline levels for multilevel factor variables are: Domain (Multiple) and International Union for Conservation of Nature (IUCN) (Unknown). Error bars mark 95% confidence intervals. Variance explained is reported as marginal $R^2$, that is, those explained by fixed factors. Asterisks (*) mark significant effects (α = 0.01). Estimated regression parameters are provided in *Supplementary file 1a*.

most important in explaining the choice of investigated species across the scientific literature (31% of explained variance) and, to an even greater extent, the number of views on Wikipedia (38%). Species-level traits explained 12% of the variance in the scientific model and 15% of the variance in the societal interest model, whereas both sets of drivers jointly contributed an additional 19 and 16%, respectively, to the two models (*Figure 3B*).

## Discussion

We found that the strongest drivers of research and societal interest are utilitarian cultural features, namely whether a species is useful and/or harmful for humans in some way (*Figure 3A*), matching previous evidence based on restricted taxonomic samples. For example, *Vardi et al., 2021* showed that in Israel, the most popular plants in terms of online representation often have some use for humans. Similarly, *Ladle et al., 2019* found that bird representation online is strongly associated with long histories of human interactions, for example, in the form of hunting or pet-keeping. From a

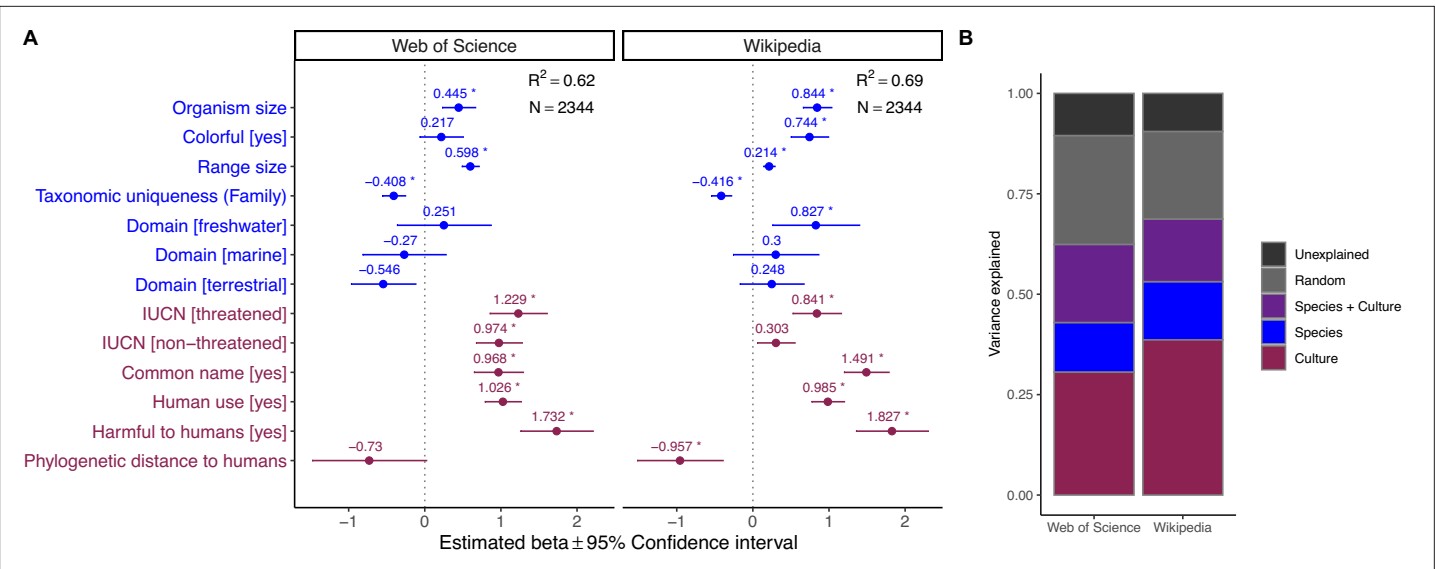

**Figure 3.** Influence of species-level traits (blue) and cultural factors (red) on the scientific and societal interest across the eukaryotic Tree of Life. (**A**) Forest plots summarize the estimated parameters based on negative binomial generalized linear mixed models (**Equation 1**). Baseline levels for multilevel factor variables are: Domain (Multiple) and IUCN (Unknown). Error bars mark 95% confidence intervals. Variance explained is reported as marginal $R^2$, that is, those explained by fixed factors. Asterisks (*) mark significant effects ($\alpha = 0.01$). Exact estimated regression parameters and p-values are provided in **Supplementary file 1a**. (**B**) Outcomes of the variance partitioning analysis, whereby we partitioned out the relative contribution of species-level traits (blue) and culture factors (red). Joint explained variance (Species + Culture) is highlighted in purple. Unexplained variance is the amount of unexplained variance after considering the contribution of random factors related to species' taxonomy and biogeographic origin (as obtained via conditional $R^2$).

cognitive standpoint, an interpretation of this relationship may be rooted in our ancestral past, when we more often relied on wildlife products and we were more frequently subject to predation and other hazards related to wildlife. Experimental evidence suggests that, even in today's society, images of dangerous animals are better able to arouse and maintain human attention (*Yorzinski et al., 2014*). Interestingly, harmfulness to humans was not a significant driver of scientific and societal interest in Tracheophyta (*Figure 4*). This result may partly be an artifact because plants dangerous to humans are those that are poisonous, but many poisonous plants are contemporary medicinal plants, making it difficult to draw a clear border between usefulness and dangerousness. This is also the case for many poisonous animals, but since vascular plants do not move, the value of their poison as a medicine might overrun our perception of them as a threat. More broadly, many species with deeply rooted histories of interactions with humans retain their importance in specific cultural contexts, particularly among indigenous peoples, and are thus more likely to remain salient nowadays. Disrupting these connections can have important biocultural consequences, and negatively affect both the species and the communities that value them (*Ladle et al., 2023*; *Reyes-García et al., 2023*).

Species with a common name also attracted more scientific and popular interest, matching previous studies (e.g., *Vardi et al., 2021*). This result should be interpreted with caution, however, because we considered only whether a species possesses an English common name. While we recognize the limitations of this approach, English was selected due to the lack of a comprehensive list of common species names in multiple languages, and because most species that are relevant in other cultural and language settings are also likely to have been attributed English common names as part of legis-lative, scientific, and other societal processes. It must also be noted that this variable entails some circularity, given that humans tend to assign common names to popular species and/or those that are relevant to humans in some way. For example, a recent study showed that across nine local villages in Mozambique, species perceived as dangerous were more likely to have a local name (*Farooq et al., 2021*). Interestingly, this speaks about the possible existence of specific interactions among different cultural traits and cultural settings which our results do not capture and could be further explored with targeted studies.

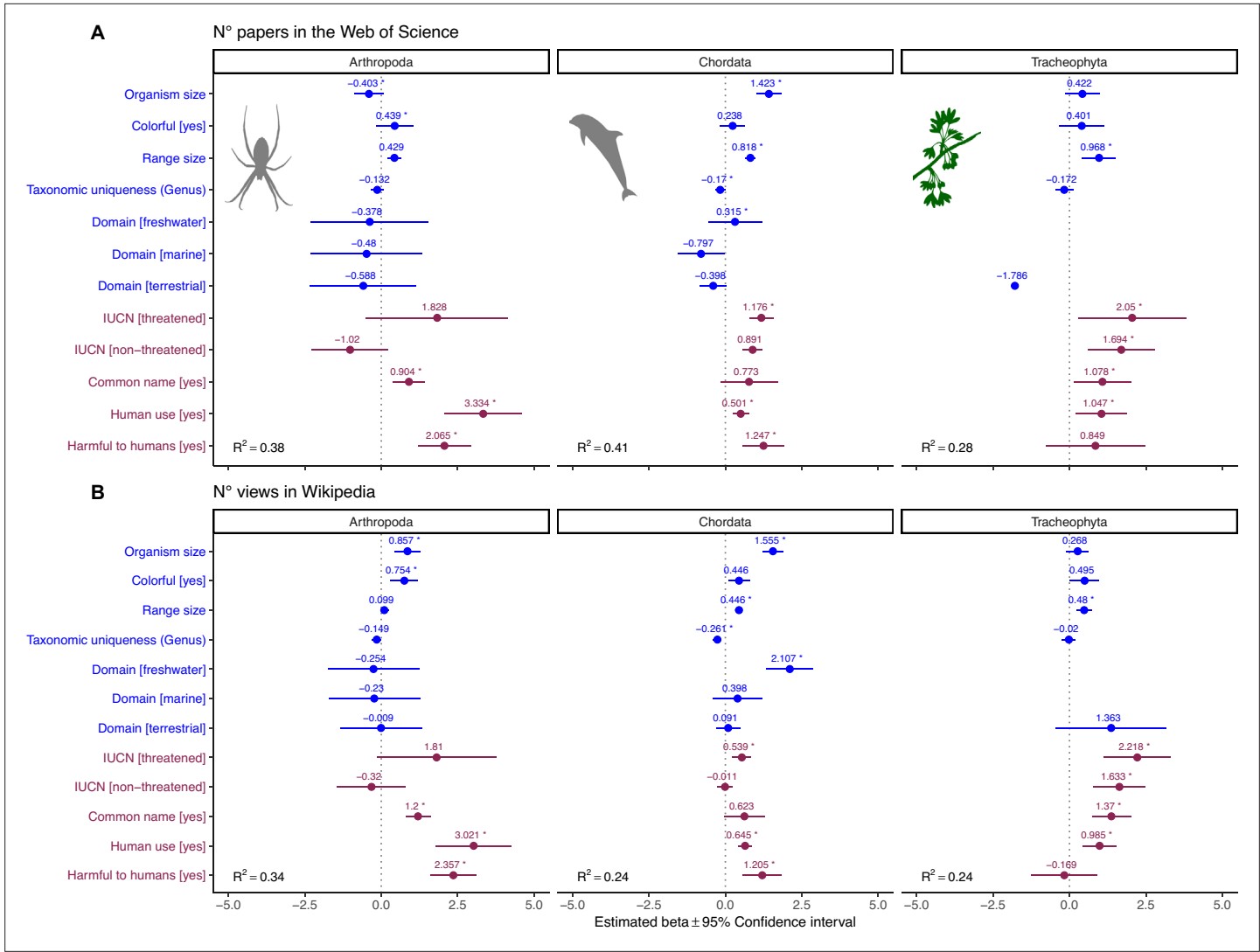

**Figure 4.** Influence of species-level traits (blue) and cultural factors (red) on scientific (**A**) and societal (**B**) interest for Arthopoda, Chordata, and Tracheophyta. Forest plots summarize the estimated parameters based on negative binomial generalized linear mixed models (*Equation 2*). Baseline levels for multilevel factor variables are: Domain (Multiple) and IUCN (Unknown). Error bars mark 95% confidence intervals. Variance explained is reported as marginal $R^2$, that is, those explained by fixed factors. Asterisks (*) mark significant effects ($\alpha$ = 0.01). Estimated regression parameters and p-values are provided in *Supplementary file 1b* (**A**) and *Supplementary file 1c* (**B**).

The positive effect of body size on scientific and societal interest suggests our attention is likely best captured by organisms with sizes similar (or larger than) our own, rather than organisms that are barely visible. Furthermore, larger species are easier to study and more detectable in the field (*Johnston et al., 2014*; *Kéry and Gregg, 2003*). Previous studies documented positive relationships between human interest and body size, for example, in different vertebrate groups (*Berti et al., 2020*; *Guedes et al., 2023*; *Ward et al., 1998*; *Żmihorski et al., 2013*) and flowering plants (*Adamo et al., 2021*), while others observed negative relationships, for example, in passerine birds (*Garrett et al., 2018*) and butterflies (*Żmihorski et al., 2013*). This hints that there may be some within-group variability that is not captured in our broad-scale analysis. However, it is worth noting that most previous studies have focused on organisms that are within the same approximate size range as humans. Indeed, when we repeated regression analyses within subsets of data corresponding to the phylum Chordata, Arthropoda, and Tracheophyta, we found that the effect of body size was not significant in Tracheophyta (*Figure 4*). While our random sample of Tracheophyta encompassed an enormous range of sizes—from a duckweed to a sequoia—it may be that attractiveness in plants is primarily controlled by other aesthetic drivers (*Adamo et al., 2021*).

Different variables reflecting both commonness and rarity contributed markedly in explaining scientific and, to a lesser extent, societal interest. The positive relationship between scientific and societal interest and geographic range size suggests a broader area of distribution could make a species accessible and visible to more people, including researchers, and thus more likely to be studied and searched for in Wikipedia. This result aligns with previous studies observing a positive correlation between proxies of species familiarity and online popularity (*Correia et al., 2016*; *Żmihorski et al., 2013*) or scientific interest (*Adamo et al., 2021*). Furthermore, taxonomically unique species often attracted more scientific and societal interest. These species may represent unique adaptations and phylogenetic distinctiveness and thus be of interest from research or conservation standpoints. Taxonomic uniqueness may also appeal and fascinate the general public, as in famous cases of the discovery of living individuals belonging to taxa previously restricted to the fossil record such as the maidenhair tree (*G. biloba* L.) or the coelacanth fish (*Latimeria chalumnae* Smith). Conservation rarity, measured as presence and status on the IUCN Red List, was also an important driver of scientific and societal interest. Concerning scientific interest, this was true regardless of the threatened status, namely both endangered and least concern species were more studied and popular across our dataset compared to unlisted species. This variable also entails a certain degree of circularity: IUCN assessments require a lot of data, making it possible to confidently assess species only when there is background information on their distribution and threats.

Finally, colorfulness and phylogenetic proximity to humans correlated exclusively with societal attention. Colorfulness is an important proxy for the aesthetic value of biodiversity (*Langlois et al., 2022*; *Santangeli et al., 2023*) and has been shown to often match cultural and economic interests—for example, it was recently shown that colorful birds and fish are more frequently targeted in wildlife trade (*Borges et al., 2022*; *Senior et al., 2022*). Phylogenetic proximity to humans seemingly correlates with a range of traits including the degree of empathy and anthropomorphism toward species. This result resonates with a recent study by *Miralles et al., 2019*, who used an online survey to assess the empathy of 3500 raters toward 52 taxa (animals, plants, and fungi) and observed a strong negative correlation between empathy scores and the divergence time separating the different taxa from *Homo sapiens* (*Miralles et al., 2019*). It is more difficult to explain the fact that freshwater-dwelling species were significantly more searched for in Wikipedia than species inhabiting multiple habitats. Speculatively, this may reflect human preference for species inhabiting habitats that are more foreign to human experience, but may also be a sampling artifact (only 103 species in the model, less than 4% of the total, were freshwater-dwelling).

The fact that subjectivity might drive scientific and societal attention toward biodiversity is not a problem per se, but, in the long run, it may bias our general understanding of life on Earth to the point of influencing policy decisions and allocation of research and conservation funding. For example, more popular species tend to receive more funding and resources for conservation efforts (*Adamo et al., 2022*; *Davies et al., 2018*; *Mammola et al., 2020*) and the allocation of protected areas has not adequately considered non-vertebrate species, as up to two-thirds of threatened insect species are not currently covered by existing protected areas (*Chowdhury et al., 2023*). This disparity in awareness may also influence species' long-term conservation prospects—a species is less likely to go extinct if humans choose to protect it. Bluntly put, it may be that we are concentrating our attention on species that humans generally consider to be useful, beautiful, or familiar, rather than species that deserve more research effort due to a higher extinction risk and/or due to the key role they play in ecosystem stability and functioning.

Excluding subjectivity when developing any research agenda is certainly challenging. However, once we are aware that utilitarian needs and emotional and familiarity factors play a key role in the development of biodiversity research globally, we can start moving toward more balanced research agendas by carefully selecting which criteria we want to focus on. Ideally, we should aim, over time, for all parameter estimates in *Figure 3A* to move toward the middle (with the possible exceptions of IUCN categories). This strategy would minimize the effect of aesthetic and cultural factors in the selection of research and conservation priorities, and can be achieved over time through a more even repartition of research and conservation funds (see, e.g., *Mammola et al., 2020* for a concrete agenda).

Global biodiversity is disappearing at an accelerating pace, not only from the physical world (*Barnosky et al., 2011*; *Cowie et al., 2022*) but also from our minds (*Donadio Linares, 2022*; *Jarić*

*et al., 2022*; *Soga and Gaston, 2016*). Given that the long-term survival of humanity is intertwined with the natural world, preserving biodiversity in all its forms and functions (including cultural awareness of it) is a central imperative of the 21st century (*Díaz and Malhi, 2022*; *Jarić et al., 2022*; *Loreau et al., 2021*). However, biodiversity goals can only be reached by ensuring a 'level-playing field' in the selection of conservation priorities, rather than looking exclusively at the most appealing branches of the Tree of Life.

## Materials and methods

### Species sampling

We carried out random stratified sampling of the eukaryotic multicellular Tree of Life (Animalia, Fungi [restricted to Agaricomycetes], and Plantae [excluding unicellular Algae]) using the Global Biodiversity Information Facility (GBIF) backbone taxonomy. To our knowledge, GBIF is the only available backbone taxonomy covering all our target groups using a congruent classification. Note that we restricted our analyses to pluricellular organisms to bypass issues with the unstable taxonomic classification of protists (*Ladle et al., 2019*; *Adl et al., 2012*; *Adl et al., 2019*) and the challenge of extracting comparable traits between unicellular and multicellular eukaryotes.

Initially, we cleaned the GBIF backbone taxonomy by subselecting only accepted names (taxonomicStatus = 'accepted'), removing subspecies and varieties (taxonRank = 'subspecies' and 'variety'), and fossil species (by removing both entirely extinct groups [e.g., Dinosauria] and single species labeled as 'Fossil_Specimen'). We chose the following criteria for the stratified random sampling:

i.  The sample was at the species level within each order of Animalia, Fungi, and Plantae (this way, we sampled all extant phyla and classes in the database).

ii. For each order, we sampled a fraction of 0.002 species. To avoid having an excessively uneven number of species among orders, we set the following thresholds:

> If the number of species in an order was comprised of between 10,001 and 50,000, we arbitrarily sampled 20 species;
> If the number of species in an order was comprised of between 50,001 and 100,000, we arbitrarily sampled 40 species; and
> If the number of species in an order was >100,000, we arbitrarily sampled 60 species.

iii. We incorporated a broader sample of tetrapods so as to reflect the typical knowledge bias ('*Institutional vertebratism*'; *Leather, 2013*). For each tetrapod order, we arbitrarily sampled 20 species. However, for small tetrapod orders with less than 10 species, we only sampled 1 species.

This random sampling procedure yielded a database consisting of 3019 species (*Figure 1—figure supplement 2C*). Despite the initial cleaning procedure of the dataset, due to the fact that some taxonomic names were not properly labeled in GBIF, 129 of the sampled names were synonyms, doubtful (*nomina dubia*), or fossils. We therefore manually inspected all records and dealt with taxonomic issues. Each expert involved in the study made decisions for their focal organisms on the invalid taxonomic names, for example, reclassifying subspecies to the species rank, replacing eventual synonyms with the currently valid name, and substituting fossils with extant species.

### Measures of scientific and societal interest

We collected data on two indicators of human attention toward species, pertaining to scientific and societal interest.

We measured scientific interest as the number of articles indexed in the *Web of Science* that refer to a given species. This is a standard quantitative estimate of research effort toward individual species (*Adamo et al., 2021*; *dos Santos et al., 2020*; *Tam et al., 2022*; *Wilson et al., 2007*). We collected data using the R package 'wosr' version 0.3.0 (*Baker, 2018*). Specifically, we queried the *Web of Science*'s Core Collection database using topic searches ('TS') and the species scientific name as the search term, and recorded the total number of references published between 1945 and the date of sampling returned by each query. The use of scientific names returns comparable results to searches using vernacular names (*Correia et al., 2017*; *Jarić et al., 2016*) but avoids common problems

associated with vernacular language queries (e.g., words with multiple meanings [homonyms] or used as brand names [theronyms]).

We measured societal interest for each species as the total number of pageviews across the languages where the species is represented on Wikipedia. Wikipedia is one of the top 10 most visited websites in the world (https://www.similarweb.com/top-websites/, accessed on February 3, 2023) and is often visited as a source of information for wildlife enthusiasts, many species containing a page in this digital encyclopedia. Wikipedia data has been widely used to explore patterns of popular interest in biodiversity, and total pageviews may be a particularly useful metric in instances where some pages have very few visits overall (*Vardi et al., 2021*). To extract the number of pageviews for each species, we first obtained the identification number of each species from the Wikidata knowledge base using the R package 'WikidataQueryServiceR' version 1.0.0 (*Popov, 2020*). We then used each species' identifier to compile a list of available Wikipedia pages for the species in any language using the same query service. Once we identified the full list of Wikipedia pages for the species, we used the R package 'pageviews' version 0.5.0 (*Keyes and Lewis, 2020*) to extract monthly user pageviews (i.e., excluding views by bots) for the period between January 1, 2016, and December 31, 2021.

## Species-level traits and associated hypotheses

To investigate the relationship between species-level traits, cultural factors, and scientific and popular interest, we selected a set of candidate variables related to species morphology, ecology, and scientific and societal preferences of humans. Extracting comparable traits across distantly related taxa is challenging (*Chichorro et al., 2022*; *Palacio et al., 2022*; *Weiss and Ray, 2019*), thus we restricted the analysis to a small number of scalable traits and kept trait resolution low (i.e., we scored most traits as categorical variables rather than on continuous scales). Importantly, to ensure cross-taxon comparability of traits, we made specific decisions on how to score traits for the different organisms (details of decisions made and sources of traits are provided in Appendix 1).

## Species-level traits

First, we extracted the average body size for each species (in mm). Size is among the most conspicuous and ubiquitous traits in ecology, relating to diverse body functions and ecological strategies (*Calder, 1996*; *Peters, 1986*). Furthermore, we expected an innate preference for large-sized species among scientists, the media, and the public alike (*Berti et al., 2020*; *Hall et al., 2011*; *Mammola et al., 2017*; *McClain et al., 2015*). We also extracted the average size of males and females to calculate sexual size dimorphism as a possible driver of interest. However, as sex-specific size values were available for <20% of species in the database, we ended up excluding this variable from analyses.

We also scored, as binary variables (Yes/No), whether individuals within a species are colorful overall (brightly colored and/or multicolored species), blue-colored (i.e., when the species has bright blue/light blue markings or overall coloration), or red-colored (when the species has bright red/purple markings or overall coloration). In the case of sexually dichromatic species, we scored these traits as 'Yes' even if only one sex displayed colorations. While there are more sophisticated ways to compute color variables (e.g., by extracting RGB pixels from standardized photographs; *Delhey et al., 2021*), this was not possible in our case since photographs were available for only 57% of the species included in our database. Given the role of aesthetics in driving human preference across diverse domains (*Hoyer and Stokburger-Sauer, 2012*), we hypothesized colorfulness to be a strong driver of attention toward biodiversity (*Langlois et al., 2022*). Furthermore, we scored red and blue patterns because these colors are known to impact people's affection, cognition, and behavior (*Elliot and Maier, 2014*). Recent studies on European plants, for example, have highlighted that species with blue/purple flowers are more frequently studied in the scientific literature (*Adamo et al., 2021*) and receive more conservation funds (*Adamo et al., 2022*).

For each species, we calculated taxonomic uniqueness as the number of species in the same family (Family uniqueness) or the number of congeneric species (Genus uniqueness). Taxonomic uniqueness may be interesting to scientists and the general public for different reasons. On the one hand, monospecific genera or families may capture divergent phylogenetic lineages defined by the presence of rare or exclusive characters (i.e., unique synapomorphies), and thus be of interest from research or conservation standpoints. On the other hand, families or genera rich in species may be useful as case

studies (e.g., to explore evolutionary radiations; *Gillespie et al., 2020*) or be of interest to the general public simply because of greater accessibility and familiarity.

We marked the main domain inhabited by each species, namely 'freshwater,' 'marine,' 'terrestrial,' or 'multiple.' Finally, we used the R package 'rgbif' version 3.7.1 (*Chamberlain et al., 2022*) to extract distribution points for each species. As in *Adamo et al., 2021*, we expressed the geographic range size of each species as the average distance between occurrence points. This measure (dispersion) is less influenced by sampling effort than commonly used proxies of range size (e.g., minimum convex polygon or the area of occupancy). Hence, it should be better suited when dealing with opportunistically collected occurrence data such as in GBIF (*Hughes et al., 2021*). Geographic range size is not only a measure of ecological commonness (*Gaston, 2011*), but also reflects species' accessibility and familiarity to scientists and the general public. Indeed, there is a tendency for humans to be more interested in wildlife species with which they have direct experience (*Ladle et al., 2016*), for example, common species that are available to us through direct experience (*Adamo et al., 2021*; *Schuetz and Johnston, 2019*). Using the GBIF coordinates, we also extracted the coordinate of the centroid of each species' range, providing a rough indication of their geographic provenance (*Figure 1—figure supplement 1*). Using the FADA Faunistic Regions database (*Balian et al., 2008*) (available at https://www.marineregions.org/; accessed on November 1, 2022), we extracted the biogeographic region in which each species occurs (Afrotropical, Antarctic, Australasian, Nearctic, Neotropical, Oriental, Pacific, and Palaearctic) based on the centroid coordinates.

## Cultural features

To express cultural knowledge and relationships between humans and wildlife, we scored, as binary variables (Yes/No), whether (i) a species has a popular name in English (*Common name*); (ii) is an established scientific model organism beyond ecology and evolution (*Model organism*); (iii) is harmful to humans in some way—for example, crop pests, invasive non-native species, species potentially dangerous to humans (large carnivores, venomous snakes, etc.) (*Harmful to human*); (iv) has any commercial and/or cultural use (used as pets, as food, for pharmaceuticals, etc.) (*Human use*); and (v) whether it has been assessed by the IUCN. Although we acknowledge that for the variables *Harmful to human* and *Human use* further subcategories could be used (e.g., crop pests, invasive, and harmful to humans may elicit different reactions and interests from a scientific and societal perspective), we decided not to split them due to sample size limitations.

We obtained divergence time (in millions of years) between each organism and *H. sapiens* from the Time Tree database (*Kumar et al., 2022*). For this, we used a modified version of the *timetree()* function in the R package 'timetree' version 1.0 (https://github.com/FranzKrah/timetree; accessed on November 8, 2021; *Krah, 2018*). First, we obtained pairwise divergence time between each taxon and *H. sapiens* by running the function at the genus rank. If the assignment failed, we ran the function iteratively up to the family rank. If still missing, we manually assigned values to the first occurring rank in Time Tree (78 taxa, 2.3% of total). We hypothesize divergence time from *H. sapiens* to be a key factor that may explain human interest in biodiversity (*Wilson, 1993*), relating to empathy and compassion towards species (*Miralles et al., 2019*) and the degree of anthropomorphism in human–organism interactions (*Servais, 2018*).

Finally, we expressed the conservation status of each species as their IUCN extinction risk, which we extracted from the IUCN Red List of Threatened species using the R package 'rredlist' version 0.7.0 (*Chamberlain, 2022*). We assigned each species to one of the following categories: Extinct (EX), Extinct in the Wild (EW), Critically Endangered (CR), Endangered (EN), Vulnerable (VU), Near Threatened (NT), Least Concern (LC), Data Deficient (DD), and Not evaluated (NE). To balance the factor levels, we later re-grouped the different categories into three levels: 'Threatened' (EX, EW, CR, EN, and VU), 'Non-Threatened' (NT and LC), and 'Unknown' (DD and NE).

## Data analysis

We used regression analyses (*Zuur et al., 2016*) to test whether there were consistent relationships between scientific (number of scientific papers) and societal (number of views in Wikipedia) interest in an organism and species-level traits and cultural features. We carried out all analyses in R version 4.1.0 (*R Development Core Team, 2021*). We used the package 'glmmTMB' version 1.1.1 for modeling (*Brooks et al., 2017*) and 'ggplot2' version 3.3.4 (*Wickham, 2016*) for visualizations. In all analyses, we

followed the general approach by *Zuur et al., 2016* for data exploration, model fitting, and validation. For data exploration, we visually inspected variable distribution, the presence of outliers, collinearity among continuous predictors (using pairwise Pearson's correlations), and the balance of factor levels (*Zuur et al., 2010*). For model validation, we used the suite of functions of the package 'performance' version 0.0.0.6 (*Lüdecke et al., 2020*) to visually inspect model residuals and evaluate overdispersion, zero-inflation, and multicollinearity. Given the large sample size of our dataset, we used a conservative approach in the identification of significance, setting an alpha level for significance at 0.01 instead of the usually accepted 0.05 (*Benjamin et al., 2018*). Furthermore, in interpreting and discussing results, we gave more relevance to explained variance and effect sizes rather than significance (*Muff et al., 2022*).

In a first set of models, we explored the role of species-level and cultural traits in explaining scientific and popular interest (dependent variables). As a result of data exploration, we log-transformed the variables *Organism size*, *Range size*, *Family uniqueness,* and *Phylogenetic distance to humans* to homogenize their distribution and minimize the effect of a few outlying observations. We dropped the categorical variable *Model organism* because it was highly unbalanced—our random sample of species across the Tree of Life only captured 15 species classified as model organisms. Likewise, the variables blue colored and red colored were unbalanced and, to a certain extent, associated with the variable *Colorful*. We used only the latter in the analyses. Finally, we scaled continuous variables to a mean of zero and a standard deviation of 1 to facilitate model convergence and interpretation of the effect sizes. We fitted the initial models assuming a Poisson error structure (suitable for count data) and a log-link function (ensuring positive fitted data). The models had the formula (in R notation):

$$y \sim Organism\ Size\ +\ Colorful\ +\ Range\ size\ +\ Domain\ +\ Taxonomic\ uniqueness\ +\ Common\ name\ +$$
$$IUCN\ +\ Humanuse\ +\ Harmful\ to\ human\ +\ Phylogenetic\ distance\ to\ humans\ +\ (1|Phylum/Class/Order)\ +(1)$$
$$(1|Biogeographic\ region)$$

where y is either the number of articles in the Web of Science (*Scientific interest*) or the number of views in Wikipedia (*Popular interest*). We introduced random factors to take into account the non-independence of observations. We accounted for taxonomic relatedness among species with a nested random intercept structure (*1 | Phylum / Class / Order*), under the assumption that closely related species should share more similar traits than would be expected from a random sample of species. Likewise, we used the random intercept structure (*1 | Biogeographic region*) under the assumption that people from the same region, including researchers, might be geographically biased in their interests, that is, share common appreciation for similar species. Both models were overdispersed (scientific interest: dispersion ratio = 47.2; Pearson's $chi^2$=109874.8; p<0.001; popular interest: dispersion ratio = 632366.5; Pearson's $chi^2$=1471516950.1; p<0.001). Therefore, we fitted new models assuming a negative binomial distribution—that is, a generalization of Poisson distribution that loosens the assumption that the variance should be equal to the mean.

Model validation for the scientific interest model revealed the existence of a highly influential observation corresponding to the Asian elephant (*Elephas maximus* L.). We therefore refitted the model removing this observation, which yielded almost identical model estimates but a better distribution of residuals versus fitted values. Also in the case of the popular interest model, there was a highly influential observation corresponding to the Mugger crocodile (*Crocodylus palustris* [Lesson]), which we removed. Model validation further revealed that the popular interest model was underfitting zeros (observed zeros: 176; predicted zeros: 95; ratio: 0.54), suggesting probable zero-inflation. Therefore, we refitted the model as a standard zero-inflated negative binomial model using the default 'NB2' parameterization implemented in 'glmmTMB' (*Hardin and Hilbe, 2007*). This substantially improved model fit (Akaike information criterion of 42727.9 vs. 42805.1). No multicollinearity affected either final model, with all variance inflation factors for covariates below 3 (*Zuur et al., 2010*).

Once the models were fitted and validated, we used variance partitioning analysis (*Borcard et al., 1992*) to estimate the relative contribution of species-level traits and cultural factors in determining the observed pattern of scientific and societal interest. We used variance explained (marginal $R^2$) to evaluate the contribution of each variable and combination of variables to the research and societal attention each species receives by partitioning their explanatory power with the R package 'modEvA' version 2.0 (*Barbosa et al., 2015*).

Next, we tested whether the importance of traits would change across the main groups of organisms by running three models within subsets of data corresponding to Arthropoda, Chordata, and Tracheophyta (i.e., the Phyla/Divisions with most observations). The structure of the models was

$$y \sim Organism\ Size + Colorful + Range\ size + Domain + Genus\ uniqueness + Common\ name + IUCN +$$
$$Humanuse + Harmful\ to\ human + (1|Class/Order) + (1|Biogeographic\ region) \tag{2}$$

The formula is essentially the same as *Equation 1*, but for the exclusion of *Phylum* from the random part (as we modeled at the Phylum/Division level) and *Phylogenetic distance to humans* from the fixed part (as we lacked enough resolution in the phylogenetic distance information within Phyla). We also used Genus uniqueness instead of Family uniqueness given that we modeled at the Phylum level. Also in this case, since Poisson models were overdispersed, we switched to a negative binomial distribution.

We also ran an analysis to understand which species-level traits drive the relative interest of scientists and the general public in different taxa. First, we used a generalized additive model to model the relationship between *Popular interest* and *Scientific interest* (*Figure 1A*). For each species, we extracted the residuals from this regression curve, whereby positive residuals indicate species with a greater popular than scientific interest, residuals close to zero indicate species with a balanced popular and scientific interest, and negative residuals indicate species with a greater scientific than popular interest (*Figure 1B*). Next, we used a Gaussian linear mixed model to model the relationship between the residuals and species-level traits. This model had the same general formula as *Equation 1*.

## Acknowledgements

The authors thank Caio Graco-Roza for helping with ggplot2. Filipe Chichorro kindly compiled traits for ants. The authors acknowledge the support of NBFC, funded by the Italian Ministry of University and Research, PNRR, Missione 4 Componente 2, 'Dalla ricerca all'impresa,' Investimento 1.4, Project CN00000033. RAC acknowledges funding from the Academy of Finland (grant agreement #348352) and the KONE Foundation (grant agreement #202101976). RS thanks the support by the Portuguese Foundation for Science and Technology (FCT) through national funds under the project MULTI-CRASH: Multi-dimensional ecological cascades triggered by an invasive species in pristine habitats (PTDC/CTA-AMB/0510/2021).

## Additional information

### Funding

| Funder | Grant reference number | Author |
|---|---|---|
| Ministero dell'Istruzione, dell'Università e della Ricerca | CN00000033 | Stefano Mammola Diego Fontaneto Matteo Chialva Martino Adamo |
| Academy of Finland | 348352 | Ricardo A Correia |
| KONE Foundation | 202101976 | Ricardo A Correia |
| Ministry of Science and Innovation | RYC2021-032594-I | David Villegas-Rios |
| Foundation for Science and Technology (FCT) | PTDC/CTA-AMB/0510/2021 | Ronaldo Sousa |
| Serbian Ministry of Science, Technological Development and Innovation | 451-03-47/2023-01/ 200178 | Dragan Antić |

The funders had no role in study design, data collection and interpretation, or the decision to submit the work for publication.

## Author contributions
Stefano Mammola, Conceptualization, Data curation, Formal analysis, Validation, Visualization, Methodology, Writing – original draft, Writing – review and editing; Martino Adamo, Matteo Chialva, Diego Fontaneto, Alejandro Martínez, Conceptualization, Data curation, Methodology, Writing – review and editing; Dragan Antić, Dan Chamberlain, Furkan Durucan, Duarte Goncalves, Iñigo Rubio-Lopez, Ronaldo Sousa, Aida Verdes, Data curation, Writing – review and editing; Jacopo Calevo, Luca Santini, Data curation, Methodology, Writing – review and editing; Tommaso Cancellario, Data curation, Visualization, Methodology, Writing – review and editing; Pedro Cardoso, Methodology, Writing – review and editing; David Villegas-Rios, Data curation, Funding acquisition, Writing – review and editing; Ricardo A Correia, Conceptualization, Data curation, Formal analysis, Methodology, Writing – original draft, Writing – review and editing

## Author ORCIDs
Stefano Mammola (ID) https://orcid.org/0000-0002-4471-9055
Martino Adamo (ID) http://orcid.org/0000-0001-7571-3505
Jacopo Calevo (ID) http://orcid.org/0000-0002-1717-2365
Matteo Chialva (ID) http://orcid.org/0000-0002-6996-6642
Furkan Durucan (ID) http://orcid.org/0000-0002-6168-2135
Diego Fontaneto (ID) http://orcid.org/0000-0002-5770-0353
Duarte Goncalves (ID) http://orcid.org/0000-0003-4299-0375
Iñigo Rubio-Lopez (ID) http://orcid.org/0000-0001-7316-7942
Ronaldo Sousa (ID) http://orcid.org/0000-0002-5961-5515
David Villegas-Rios (ID) http://orcid.org/0000-0001-5660-5322
Aida Verdes (ID) http://orcid.org/0000-0002-9193-9253

Reviewer #1 (Public Review): https://doi.org/10.7554/eLife.88251.3.sa1
Reviewer #2 (Public Review): https://doi.org/10.7554/eLife.88251.3.sa2
Author Response https://doi.org/10.7554/eLife.88251.3.sa3

## Additional files

### Supplementary files
• Supplementary file 1. Estimated parameters for the regression models. (**a**) Estimated regression parameters for the full models. (**b**) Estimated regression parameters for the subset models modeling the number of papers in the Web of Science. (**c**) Estimated regression parameters for the subset models modeling the number of views in Wikipedia.

• MDAR checklist

### Data availability
The database used in the analyses is available in Figshare (https://doi.org/10.6084/m9.figshare.22731440.v1). The R code to generate analyses and figures is available on GitHub (https://github.com/StefanoMammola/Mammola_et_al_ToL_research_interest copy archived at *Mammola, 2023*).

The following dataset was generated:

| Author(s) | Year | Dataset title | Dataset URL | Database and Identifier |
| --- | --- | --- | --- | --- |
| Mammola S, Adamo M, Antić D, Calevo J, Cancellario T, Cardoso P, Chamberlain DE, Chialva M, Durucan F, Fontaneto D, Gonçalves D, Martínez A, Santini L, Rubio-López I, Sousa R, Villegas-Ríos D, Verdes A, Correia RA | 2023 | Data for "Drivers of species knowledge across the Tree of Life" | https://doi.org/10.6084/m9.figshare.22731440.v1 | figshare, 10.6084/m9.figshare.22731440.v1 |

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

## Appendix 1

### Additional text

Information on trait extraction for the different groups. Note that the grouping of taxa is not always at the same level of Linnaean rank, but rather it reflects the groups that were tackled by the different experts (see author list).

### Acanthocephala

We derived data on specific traits from the original description (or redescription) of each. Average size is the total body length. Information on size according to sex is unknown or not reported for most taxa.

### Annelida

We derived data on specific traits from the original description (or redescription) of each. Average size is the total body length. Information on size according to sex is unknown or not reported for most taxa.

## Arthropoda

### Arachnida

For Araneae, we derived specific traits from the original description of each species (*World Spider Catalog, 2022*), and complemented information by searching relevant papers and information in Google and Google Scholar. Male and female size is the total body length (prosoma + opisthosoma), including chelicerae. Habitat is simply the habitat of the type locality unless additional information was available in ecological/faunistic papers. For other arachnid orders, we derived specific traits from the original descriptions of the species (World Arachnida Catalog; https://wac.nmbe.ch/), and complemented information by searching relevant papers and information in Google and Google Scholar. For Acari, total body length is given after excluding the gnathosoma. For oribatid mites, total body size is given as the length of notogaster + prodorsum. For classes with a tail or telson (e.g., scorpions and microwhip scorpions), body length is the sum of prosoma + opisthosoma + tail/telson.

### Insecta

We retrieved the imaginal phase trait information by searching relevant papers and information in Google, Google Scholar, the Biodiversity Heritage Library, and in entomological books. Male and female size is the total body length (head + thorax + abdomen). We reported only the mean body length when no information about the sex was available. We classified the domain as the natural environment of the type locality unless additional information was available in supplementary literature.

### Merostomata

We derived specific traits from the original description of each species when available and from additional relevant literature, including taxonomic reviews and monographs. Information on size according to sex is unknown or not reported for most taxa. Habitat corresponds to that of the type locality unless additional information was available in the supplementary literature.

### Myriapoda

Specific traits for each species were derived from original descriptions and/or additional relevant literature. The majority of the information about the literature was found on Millibase (https://www.millibase.org/aphia.php?p=search). Body size is the distance between the anterior margin of the head and the posterior margin of the last body segment, excluding the antennae and/or the last pair of legs. Original descriptions and/or additional relevant literature were used to extract habitat information.

### Entognatha

We derived specific traits from the original description of each species when available and from additional relevant literature, including taxonomic reviews and monographs. Information on size according to sex is unknown or not reported for most taxa. Habitat corresponds to that of the type locality unless additional information was available in the supplementary literature.

## Pygnogonida

We derived specific traits from the original description of each species when available and from additional relevant literature, including taxonomic reviews and monographs. Information on size according to sex is unknown or not reported for most taxa. Habitat corresponds to that of the type locality unless additional information was available in the supplementary literature.

## Basidiomycota

We retrieved data through web-based searches using the Google search engine. When available, we assessed the original species description using the Biodiversity Heritage Library (https://www.biodiversitylibrary.org/). We retrieved habitat information from online searches and compared it with GBIF data. We ranked species as harmful if known to be poisonous or toxic to humans. We considered the pileus/fruiting body diameter as the average size.

## Bryozoa

We derived data on specific traits from the original description of each species when available and from additional relevant literature, including taxonomic reviews, monographs, and online databases (WORMS; *Horton, 2021*). Average size is the mean zooid length. Information on size according to sex is unknown or not reported for most taxa. Habitat corresponds to that of the type locality unless additional information was available in the supplementary literature.

## Chaetognata

We derived data on specific traits from the original description of each species (when available) and from additional relevant literature, including taxonomic reviews, monographs, and online databases (WORMS; *Horton, 2021*). Average size is the total body length. Information on size according to sex is unknown or not reported for most taxa. Habitat corresponds to that of the type locality unless additional information was available in the supplementary literature.

## Chordata

### Amphibia

We retrieved the average size and domain for most species from *Oliveira et al., 2017*. The remaining length values were mostly taken from the database AmphibiaWeb, either English or French language accounts, and very few had to be retrieved from the original description or other literature. We considered Amphibian species as harmful if they were classified as invasive/pests. No venomous urodele or anuran species were present in the list. Caecilian secretion inoculation systems were only very recently described and their toxicity is still unclear, but nevertheless these animals were not considered harmful since no envenomations are known. Harm due to improper ingestion or preparation (e.g., amphibians with toxic secretions) was also not considered since every food item has the potential to be harmful. Species solely harmful to human pets were not considered as harmful to humans.

### Aves

For each species, we used data from various online sources, especially https://www.oiseaux.net/, https://birdsoftheworld.org/bow/home, and the IUCN Red List (*IUCN, 2020*), to find specific traits. We derived additional information from various books (*Cramp and Perrins, 1994*; *Sibley, 2000*; *Stevenson and Fanshawe, 2002*) and from scientific articles located through searches in *Web of Science*. Details of the measurements of body size were rarely given, but the standard measurement is bill tip to tail tip of a laid-out specimen, which we assumed unless specified otherwise. Information on human use, domain, habitat, and trophic role was based primarily on that given in the IUCN Red List (*IUCN, 2020*); otherwise, we used scientific publications. We selected red or blue colorations based on 'strikingness' rather than extent (e.g., a black bird with a small bright red crown would be scored as red colored) assessed from online images (images were available for all selected bird species).

### Fish (Actinopterygii, Chondrichthyes, Hyperoartia, Myxini, and Sarcopterigii)

We collected data on fish species from these five classes from several online databases such as Fishbase, the IUCN red list (*IUCN, 2020*), and WORMS (*Horton, 2021*). We gathered data on domain, reproductive habitat, and trophic role in the specialized literature when not available on

the online databases. Body size refers to the maximum body length. The field 'human use' includes both species used for human consumption and popular species for aquarists. In the latter case, we checked specialized webpages to determine if a species was regularly used as an aquarium fish.

## Mammalia

We compiled data on mammals from a variety of sources including the *Handbook of Mammals of the World* (body size measurements, Diet), AnimalDiversity web (body size measurements, human use), and the IUCN Red List (*IUCN, 2020*) archive (habitat preferences and human use). We expressed body size as head-body length in all cases except cetaceans for which size measures are expressed as total length (including the tail length).

## Reptilia

We retrieved squamate body length (maximum SVL) retrieved from *Slavenko et al., 2019*. Data comprise SVL for lizards and amphisbaenians, while for snakes most of the data (ca. 90%) comprise total lengths (TL) and the rest SVL. Crocodilian body length values were obtained from *Trutnau and Sommerlad, 2006*. Turtle carapace lengths were obtained from *Itescu et al., 2014*. It should be noted that there is a reptile-standardized mass size-value, based on clade-specific allometric equations (updated in *Slavenko et al., 2019*; *Slavenko et al., 2016*), but that was not used to keep coherence with other taxa in this study.

## Cnidaria

We derived data on specific traits from the original description of each species when available and from additional relevant literature, including taxonomic reviews, monographs, and online databases (WORMS; *Horton, 2021*). Average size is the total body length of a specimen. The size of a colonial species correlates to the overall size of the colony since this is what is recognized in the field. Information on size according to sex is unknown or not reported for most taxa. Habitat corresponds to that of the type locality or to the habitat stated in the review monographs unless additional information was available in the supplementary literature.

## Ctenophora

We derived data on specific traits from the original description of each species when available and from additional relevant literature, including taxonomic reviews and monographs (WORMS; *Horton, 2021*). Average size is the body length measured from the aboral pole to the tip of the mouth. Information on size according to sex is unknown or not reported for most taxa. Habitat corresponds to that of the type locality unless additional information was available in the supplementary literature.

## Cycliophora

We derived data on specific traits from the original description of each species and from additional relevant literature. Average size is the total body length. Information on size according to sex is unknown or not reported. Habitat corresponds to that of the type locality unless additional information was available in the supplementary literature.

## Echinodermata

We derived specific traits from the original description of each species when available and from additional relevant literature, including taxonomic reviews, monographs, and online databases (WORMS; *Horton, 2021*). Average size is body length for Holothuroidea, body diameter for Echinoidea and Asteroidea, arm length for Crinoidea, and disc diameter measured from the distal edge of the radial shields to the edge of the opposite interradial for Ophiuroidea. Information on size according to sex is unknown or not reported for most taxa. Habitat corresponds to that of the type locality unless additional information was available in the supplementary literature.

## Hemichordata

We derived data on specific traits from the original description of each species when available and from additional relevant literature, including taxonomic reviews, monographs, and online databases (WORMS; *Horton, 2021*). Average size is the total body length. Information on size according to sex is unknown or not reported for most taxa. Habitat corresponds to that of the type locality unless additional information was available in the supplementary literature.

## Kinorhyncha

We derived data on specific traits from the original description of each species when available and from additional relevant literature, including taxonomic reviews, monographs, and online databases (WORMS; *Horton, 2021*) Average size is the total body length. Information on size according to sex is unknown or not reported for most taxa. Habitat corresponds to that of the type locality unless additional information was available in the supplementary literature.

## Loricifera

We derived data on specific traits from the original description of each species when available and from additional relevant literature, including taxonomic reviews, monographs, and online databases (WORMS; *Horton, 2021*). Average size is the body length excluding the mouth cone. Information on size according to sex is unknown or not reported for most taxa. Habitat corresponds to that of the type locality unless additional information was available in the supplementary literature.

## Mollusca

We derived data on specific traits from the original description of each species when available and from additional relevant literature, including taxonomic reviews, monographs, and online databases (MolluscaBase; *MolluscaBase eds, 2023*). Average size is the total body length. Information on size according to sex is unknown or not reported for most taxa. Habitat corresponds to that of the type locality unless additional information was available in the supplementary literature .

## Nemertea

We derived specific traits from the original description of each species when available and from additional relevant literature, including taxonomic reviews, monographs, and online databases (WORMS; *Horton, 2021*). Average size is the total body length. Information on size according to sex is unknown or not reported for most taxa. Habitat corresponds to that of the type locality unless additional information was available in the supplementary literature.

## Nematoda

We derived data on specific traits from the original description of each species when available and from additional relevant literature, including taxonomic reviews and monographs. Average size is the total body length. Information on size according to sex is unknown or not reported for most taxa. Habitat corresponds to that of the type locality or to the habitat stated in the review monographs unless additional information was available in the supplementary literature

## Nematomorpha

We derived specific traits from the original description of each species when available and from additional relevant literature, including taxonomic reviews, monographs, and online databases (WORMS; *Horton, 2021*). Average size is the total body length. Information on size according to sex is unknown or not reported for most taxa. Habitat corresponds to that of the type locality unless additional information was available in the supplementary literature.

## Platyhelminthes

We derived data on specific traits from the original description of each species when available and from additional relevant literature, including taxonomic reviews, monographs, and online databases (*Gibson et al., 2014*; *Horton, 2021*). Average size is the total body length. Information on size according to sex is unknown or not reported for most taxa. Habitat for parasitic species corresponds either to the habitat of the animal they parasite or (when known) the domain where the larval free-living stages inhabit.

## Porifera

We derived specific traits from the original description of each species when available and from additional relevant literature, including taxonomic reviews, monographs, and online databases (WORMS; *Horton, 2021*). Average size is the total body length of a specimen. Information on size according to sex is unknown or not reported for most taxa. Habitat corresponds to that of the type

locality or to the habitat stated in the review monographs unless additional information was available in the supplementary literature.

## Priapulida

We derived specific traits from the original description of each species when available and from additional relevant literature, including taxonomic reviews, monographs, and online databases (WORMS; *Horton, 2021*). Average size is the body length excluding the tail. Information on size according to sex is unknown or not reported for most taxa. Habitat corresponds to that of the type locality unless additional information was available in the supplementary literature.

## Rotifera

We derived specific traits from the original descriptions and published literature, especially guides and taxonomic reviews (extracted from *Jersabek and Leitner, 2013*). Size is the total body length when fully extended, which is not available for the yet unknown males for most species.

## Tardigrada

We derived specific traits from the original description of each species when available and from additional relevant literature, including taxonomic reviews, monographs, and online databases (WORMS; *Horton, 2021*). Average size is the total body length. Information on size according to sex is unknown or not reported for most taxa. Habitat corresponds to that of the type locality unless additional information was available in the supplementary literature.

## Viridiplantae (Anthocerotophyta, Bryophyta, Marchantiophyta, Tracheophyta) and Rhodophyta

We retrieved data from web-based searches, using the species' binomial in the Google search engine. Where available, we obtained data from online databases and websites encompassing the main floras worldwide. If no information was retrieved, we accessed the original species description using the Biodiversity Heritage Library (https://www.biodiversitylibrary.org/) (if available). Given the differences in taxonomies adopted in GBIF and the different mined sources, when searches using GBIF taxonomy did not return any useful data, we used synonyms. We retrieved habitat information from accessed sources (both web-based searches and literature) when available, and compared with available information in GBIF. We checked species usefulness for humans using the 'Useful Tropical Plant' (http://tropical.theferns.info/) and the 'Pl@ntUse' (https://uses.plantnet-project.org/) databases or by web searches looking for uses as raw materials, food, or other purposes such as cosmetics, medicine, sources of phytochemicals, or domesticated plants. We classified species as harmful if they are known to be poisonous or toxic to humans. We calculated the average size using the maximum and minimum stem height for herbs, or overall size for trees and shrubs. When not available, we only considered the maximum size. For algae and bryophytes, we approximated size as the thallus height.

