## [Editor Report · eLife assessment]

With a carefully collected dataset and **compelling** analyses, this **fundamental** manuscript demonstrates detailed links between societal and academic interest and natural species across the globe. In doing so, the authors reveal biases that may be diminishing our abilities to care for the species on our planet that may need our care the most. While some parts of this manuscript reflect previously published work, the authors are commended for putting all the puzzle pieces together for the first time. Their work highlights our uneven knowledge of biodiversity and its potential causes.

---

## [Referee Report · Reviewer #1 (Public Review)]

In the "Drivers of species knowledge across the Tree of Life", Mammola and collaborators explore the determinants of scientific and societal interest across very wide taxonomic and spatial scales. Their work highlights our uneven knowledge of biodiversity and its potential causes.

---

## [Referee Report · Reviewer #2 (Public Review)]

Using standard and widely used tools, the author revealed the factors (cultural, phenotypic, phylogenetic, etc.) shaping societal and scientific interest in natural species around the globe. The strength of this ms (and the authors) lies in its command of the available literature, database and variable management and analysis, and its solid discussion. The authors thus achieved a manuscript that was pleasant to read.

While I agree that doing a global study requires losing details of local patterns, maybe this is exactly the biggest shortcoming of the manuscript, oblivious to how different cultures (compare USA to PNG, for example) are reflected in these global patterns.

---

## [Author Response]

The following is the authors’ response to the original reviews.

**Reviewer #1 (Public Review)**
Overall, I quite enjoyed reading the manuscript and found it very well-structured and organized. I congratulate the authors for building this nice research. I do have a few major points to raise, but probably they would not affect the general message of the manuscript.

Thank you for taking the time to review our manuscript and the positive feedback. Following your suggestions, we have corrected some mistakes and added clarifications and a few of the suggested quality checks on the models. However, we decided not to run new analyses as: (i) we believe there would be minor changes to the general message of the manuscript; and (ii) while some suggested analyses are compelling, they are difficult to implement for different reasons or are outside the scope of the paper (clarified below).

I was confused about how IUCN data were used. The IUCN predictors are not mentioned in the model equations presented in the manuscript, but their effect size is reported in Figure 2.

Thank you for highlighting this issue. This was a typo: we forgot to mention the variable in both equations 1 and 2. Changed accordingly.

In the manuscript Methods, it is said that IUCN data was classified into 3 categories. I believe there was a mix of mechanisms in measuring it this way since at least two processes might be underlying IUCN data. First, one can inspect whether there is an effect on "scientific/societal interest" for assessed vs non-assessed species. This would not have any relationship with the assessed status itself. Assessed species are any with LC, NT, VU, EN, CR, EW, EX statuses, whereas non-assessed species might include DD and NE. Second, one may observe an effect of threat status itself, with threatened species being more researched than non-threatened species, this would only be possible for assessed species, although there are methods out there to impute missing statuses. By inspecting Figure 2, I got the feeling that only the second option was explored, but this would need to be confirmed.

We couldn’t test the effect of single categories (LC, NT, VU, EN, CR, EW, EX) because observations within factor levels were unbalanced. So, we re-grouped the different categories into three levels: “Threatened” (EX, EW, CR, EN and VU), “Non-Threatened” (NT and LC), and “Unknown” (DD and NE) and only tested this variable (your second option). Note that the effect size of the level Unknown is not shown in Figure 2 as this is the reference category. This is clarified in the caption of Figure 2.

In Figure 2, I was confused about the presence of three categories of domain. In the text, it states that four categories have been used. I believe these domains are non-mutually exclusive, that's why there is a fourth category. Would it not be better to assess the influence of domain through three dummy variables (terrestrial, marine, freshwater), where multiple presences (1's) would indicate the "multiple" category?

We opted for a categorical variable (rather than a dummy) to have the same number of variables in the two groups (‘species’ vs ‘culture’). This is needed for the variance partitioning analysis (VPA), because an unbalanced number of variables in one group of a VPA can artificially inflate R2 (see, e.g., this source: https://www.davidzeleny.net/anadat-r/doku.php/en:varpart).As for Figure 2, the level “Multiple”, being the reference category, is not shown. This is clarified in the caption: “Baseline levels for multilevel factor variables are: Domain [Multiple]”.

At present, I felt that the spatial components of your data were unexplored. Since you have centroids representing species distribution, it could be interesting to explore the presence of the species within protected areas or biodiversity hotspots. That might be something triggering at least scientific interest. Also, one can derive information about the major habitat of species occurrence (either using IUCN Major Habitat classification) or extracting overlap of species centroids with WWF biomes (e.g., simplified to just forested vs non-forested habitats; https://ecoregions.appspot.com/). Another point very common to research exploring biodiversity shortfalls is the proximity to research institutions (https://doi.org/10.1111/2041-210X.13152). And since societal interest is also being explored, what about the proximity to major cities (doi:10.1038/nature25181). Finally, other metrics derived from species centroids could inform "tropicality", if the species is tropical or not. Most often, the tropics species are neglected in comparison with those occurring in temperate regions.

We thank the reviewer for this suggestion, and we are aware that there are important spatial drivers of interest as highlighted in earlier research. Indeed, the spatial aspects of the data were somewhat underexplored as a deliberate choice because we hope to carry out additional work to explore these aspects in more detail. Nevertheless, we included the centroid of each species range as a broad proxy of its distribution, to help explore, for example, the role of species latitudinal distribution in driving interest metrics. We have also considered the suggestions provided as additional analyses, but we find these may be challenging to implement with the current data for a few reasons. First, each species centroid was calculated based on GBIF occurrences and therefore represents the midpoint of all locations, but not necessarily an area that is known to be occupied by the species. Using the centroid to assess whether a species is located in a given biome or within protected areas using this approach would therefore be potentially misleading (for example, for some terrestrial species it may fall in the sea, and vice versa). Also, for the same reasons, taking the centroid to estimate the species accessibility or proximity to research institutions may be misleading. We find that while important, these spatial aspects require a more nuanced approach to be explored in detail.

I was also thinking about the influence of time on the models. Species described long ago are often more known to people and scientists and had more "time" to be researched. Although metrics of societal interest were restricted to the last decade here, that does not necessarily mean that peoples' interest is not affected by their accumulated experiences. Similar reasoning applies to scientific interests, which have a lengthier time frame (~80 years). That said, the year of description or time since description could be added to capture some metric of time.

This is a good point, which we discussed prior to running the analysis. Indeed, there is evidence that such accumulated experiences can drive species interest as our own research has also previously highlighted (e.g. see Ladle et al. 2017 doi: 10.1002/pan3.10053). However, we felt that comparing the date of description as a proxy of accumulated human experiences with species was only fair within major biological groups and not between them. This is because taxonomic practices, definitions, and methods vary widely between biological groups. We therefore decided not to include time since description as a variable driving the measures of scientific and societal interest in this work. Nevertheless, we recognize the importance of the history of such experiences in driving human interest in species, and the consequences emerging from the loss of such links, and have thus included a brief discussion of this topic in the manuscript (see lines 177-182).

Model residuals could be checked for phylogenetic or spatial autocorrelation. I am aware there is no phylogenetic tree used, but the hierarchical taxonomy could be used (Phylum / Class / Order / Family / Genus) as a proxy for phylogenetic relationship.

We agree. Indeed, the hierarchical taxonomy was already included as a random factor (Phylum / Class / Order) in eq. 1. Note that we excluded Family and Genus from the random structure because in most Phyla a single genus and family has been sampled (as well as due to model convergence problems).

Concerning the spatial autocorrelation, one could check whether model residuals and their respective coordinate centroids of each species range. It is stated that GLMM has been used to avoid these non-independence issues, but it would be interesting to check whether residuals remained free of them.

Good suggestion, although the use of centroids may not be the most appropriate since it is only a rough approximation of each species distribution (see previous answer). Still, out of curiosity, we checked whether the random factor on biogeography was enough to capture residual spatial autocorrelation in the models. For this, we used the R package DHARMa, which performs a Moran's I test for distance-based autocorrelation. Given that some coordinates were duplicated, we grouped residuals by biogeographic regions (DHARMa requires all coordinates to be unique). Neither the Web of Science nor the Wikipedia models had spatial autocorrelation in the residuals:

Web of Science model: observed = –0.20482, expected = –0.14286, sd = 0.10682, p-value = 0.561

Wikipedia model: observed = –0.180820, expected = –0.142857, sd = 0.055513, p-value = 0.4941

A last point, it would be interesting to provide some sort of inset plots, such as barplots or donut plots (within the current plots), showing the proportion of species with respect to major clades and biogeographical regions.

This is a good suggestion, but we couldn’t find a good way to show this as an inset. We added a barchart showing the number of species in each Phyla/Division in the supplementary materials (Figures S2C). As for the proportion of species in each region, we thought it would be redundant with Figure S1 (summarizing spatial information in sampled species).

**Reviewer #2 (Public Review):**
Using standard and widely used tools, the authors revealed the factors (cultural, phenotypic, phylogenetic, etc.) shaping societal and scientific interest in natural species around the globe. The strength of this manuscript (and the authors') lies in its command of the available literature, database and variable management and analysis, and its solid discussion. The authors thus achieved a manuscript that was pleasant to read.

Thank you for taking the time to review our manuscript and the positive feedback.

While I agree that doing a global study requires losing details of local patterns, maybe this is exactly the biggest shortcoming of the manuscript, oblivious to how different cultures (compare USA to PNG, for example) are reflected in these global patterns.Related to this previous point, my only other comment is about using English as a reference of societal interest (i.e., the presence of a common name in English). While English may be widespread in Academia, it is still not that common in other societal circles, especially those not using Wikipedia for lack of internet access.

We acknowledge the limitation of this choice, as well as our limited capacity to represent specific cultural contexts with our approach. Our decision to consider only the existence of English common names as a variable was partly driven by practical reasons, and partly by the very factors the reviewer highlights. Indeed, many cultures, communities and social circles do not use English frequently and also do not use the internet frequently. One consequence of this is also that the information compiled for species in other languages is more restricted than that available in English, including the existence of vernacular names. In languages other than English, it may even be the case that several common language names exist in reference to the same species, and this number may be an even better reflection of their cultural importance, but sadly this information is not comprehensively indexed across languages and biological groups which prevented us from considering it. On the other hand, most species have been attributed English common names as part of legislative, scientific and other societal processes, and it is therefore likely that if they are important in any specific cultural setting, they will probably also have a vernacular English language name. Ultimately, while we recognize the potential limitations of this decision, we felt that considering English common names was the simplest and less biassed approach to represent the degree with which a species is individually recognized nowadays. We now better expose the reasons for the decision to consider only English common names, and the limitations associated with it in the manuscript (lines 178-193).